# Four-Level Quasi-Nested Inverter Topology for Single-Phase Applications

**Carlos A. Reusser** [1,*,†] and **Hector Young** [2,†]

1    School of Electrical Engineering, Pontificia Universidad Catolica de Valparaiso, Valparaiso 2340000, Chile
2    Department of Electrical Engineering, Universidad de La Frontera, Temuco 4811230, Chile; hector.young@ufrontera.cl
*    Correspondence: carlos.reusser@pucv.cl
†    These authors contributed equally to this work.

**Abstract:** In this paper, a novel four-level single-phase multilevel converter is introduced, consisting of six active switches arranged in a quasi-nested configuration. The proposed topology synthesizes its output voltage levels with respect to a floating neutral point, using four cascaded capacitors with identical voltage levels. The proposed converter contains a reduced number of components compared to the neutral point clamped (NPC) or active-NPC topologies (ANPC) for the same number of output voltage levels, since it does not require diode or active switch clamping to a neutral point. Moreover, no floating capacitors with asymmetric voltage levels are employed, thereby simplifying the capacitor voltage balancing. The switching operation principles, modulation technique and control scheme for supplying a single-phase resistive-inductive load are addressed in detail. The proposed four-level inverter allows generating an additional output voltage level with the same semiconductor count as conventional three-level inverters such as NPC and ANPC which allows a superior waveform quality, with a $THD_v$ reduction of 32.69% in comparison the clamped inverters. Experimental tests carried out in a laboratory-scale setup verify the feasibility of the proposed topology.

**Keywords:** multilevel inverter; quasi-Nested topology; reduced active switches topology; level-shifted PWM; DC-link voltage self-balancing; single-phase applications; proportional multi resonant controller

## 1. Introduction

Nowadays, multilevel inverters (MLI) have become the main solution for applications where high power quality and dynamic performance are required, particularly in medium- and high-power systems [1,2]. The main reasons for their success are the reduced blocking voltage rating of the semiconductor switches, less conduction losses, and lower harmonic content, which lead to increased efficiency and reduction of manufacturing costs [3,4]. Conventional MLI topologies are primarily the cascaded H-Bridge (CHB), the neutral-point clamped (NPC) and the flying capacitor (FC) configurations [5–8].

Even though conventional MLI topologies are considered as today's standard because of their technological maturity, they are subject to intrinsic limitations that have motivated the design of alternative MLI configurations [9,10]. In the case of NPC, there is an important increase in its component count for configurations with higher number of levels. For example, the 3-level NPC basic cell shown in Figure 1a has 4 active switches and 2 clamping diodes, whereas a 4-level configuration requires 6 of each devices and for 5-levels the component count reaches 8 active switches and 12 diodes. An additional drawback of the NPC topology is the requirement of complex capacitor voltage-balancing schemes [11]. As for the CHB, depicted in Figure 1b, higher number of levels with this topology requires isolated DC supplies, which increase the converter cost and complexity. The FC, shown in Figure 1c has the drawback of a high number of bulky DC capacitors which require a dedicated pre-charge strategy that adds complexity to the system [11,12].

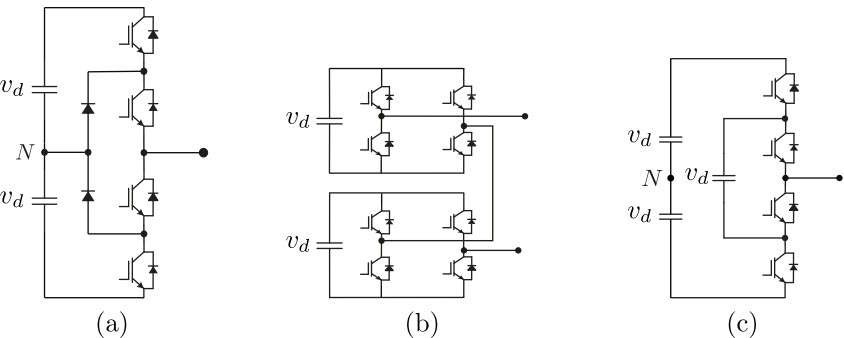

**Figure 1.** Conventional multilevel inverters (MLI) topologies: (**a**) neutral-point clamped (NPC), (**b**) cascaded H-bridge, (**c**) flying-capacitor.

MLIs find their primary field of application in high-power medium-voltage systems; however, their key advantages over two-level inverters have made them attractive for single-phase low-voltage applications. The main advantages of MLIs are waveforms with lower harmonic content and less total harmonic distortion (THD), which makes possible a reduction of filter sizes, as well as reduced stress due to voltage rate of change ($dv/dt$) and electromagnetic interference [13]. A relevant application of single-phase inverters is the transformerless integration of photo-voltaic (PV) power generation systems [14,15], where MLI topologies have been proposed recently [16–20]. Other applications of single-phase MLIs can be found in active power filters [21] and electric motor drives [22,23].

In recent years, research efforts in the field of MLI topologies have been focusing on overcoming some of the drawbacks of the conventional converters, by developing features such as single DC-source operation, avoidance of flying capacitors and reduced number and complexity of semiconductor switches [1,10,24]. In this context, the four-level double-star converter topology was developed originally by Kouro et al. [25] and has been studied for applications such as grid-connected PV energy systems [26] or back-to-back connection in wind energy systems [27]. Among the advantages of that topology are the availability of four output voltage levels using 6 active switches per phase without any clamping elements, whereas an NPC would require the same number of active switches and 4 clamping diodes for the same number of voltage levels [11]. Moreover, the four-level double-star converter does not require flying capacitors, which avoids the complications of maintaining the capacitor voltage levels [12].

Motivated by the aforementioned advantages of the topology [25], and the fact that NPC and ANPC topologies with four levels or more, so as the FC topology, are not suitable for single-phase applications, in this paper we propose a novel single-phase MLI topology derived from the four-level double-star converter called four-level quasi-nested (4L-QN) inverter. The converter topology, operating principles, switching states and modulation scheme are analyzed in detail. Simulation results and experimental validation are provided under different operational conditions, to analyse the potential of the proposed topology. The main contributions of this research are summarized as follows:

- The hardware modifications required for single-phase implementation of the 4-level double star MLI are detailed.
- A suitable carrier-based modulation scheme is proposed and tested in order to guarantee accurate capacitor voltage balancing.
- An experimental validation of the 4L-QN topology is presented for the first time.

The remainder of this paper is organized as follows: in Section 2 the proposed converter topology, switching states and implemented modulation scheme are discussed. Section 3 presents the main control objectives and finally in Section 4 simulation results and experimental verification of the proposed multilevel topology control are presented and discussed.

## 2. Converter Topology

The proposed 4L-QN is shown in Figure 2. The topology consists of four series-connected DC capacitors, tied to a common neutral point N. An arrangement of six active switches, IGBTs or MOSFETs may be used, thus IGBT semiconductor structure could be replaced by a reverse blocking MOSFETs for a high-voltage [28] at high-switching frequency operation [29]. The arrangement of active switches is based on individual H-Bridge single cells. The upper and lower output voltage levels constitute a single half-cell whose output terminals are labelled *a* and *b*, respectively. Each half-cell output point *a* and *b* is connected to a full leg consisting of two semiconductors in series in their corresponding collector and emitter terminals. The single-phase output terminal is denoted by *x*, and the load is connected between the output port and the neutral point *N*.

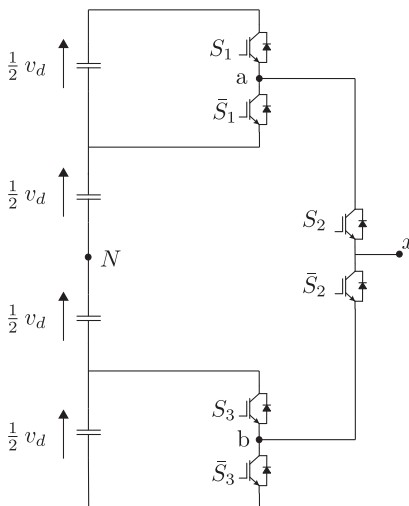

**Figure 2.** Four level quasi-nested (4L-QN) inverter topology cell.

As shown in Figure 2, the proposed converter topology is similar to a three-level NPC converter (3L-NPC) single cell topology, but without requiring clamping diodes or switches as in the case of the active-NPC converter (ANPC) topology. These topologies are classified under the concept of nested arrangement, because the central point of each leg corresponding to a fundamental cell, is connected to each other.

In comparison to the nested concept, the proposed topology does not connect each leg central point (leg output point) with each other, but the inner legs central points are connected to a single output leg, thus constituting a quasi-nested topology. The most remarkable features of this topology are the following:

- It is capable of single or three-phase operation. The latter requires using three parallel cells and a single DC-link with three capacitors and no clamping point.
- Four voltage levels: one more than the 3L-ANPC using the same number of active switches and one more level than the 3L-NPC with the same number of semiconductor switches.
- No need of clamping diodes or active switches like in the 3L-ANPC, 3L-NPC topologies.
- No need of flying capacitors with asymmetric voltage levels, thereby avoiding the requirement of a dedicated pre-charge strategy.
- The inner switches ($S_1$, $\overline{S_1}$, $S_3$, $\overline{S_3}$) blocking voltage stress is $\frac{1}{4}$ of the nominal DC-voltage $v_d$, representing an improvement with respect to nested topologies, like 3L-NPC, 3L-ANPC and H-Bridge, whose blocking voltage stress is $\frac{1}{2}$ of the DC-voltage. On the other hand, the blocking voltage stress of the outer active switches ($S_2$, $\overline{S_2}$) is $\frac{3}{4}v_d$, exceeding that of the 3L-NPC, 3L-ANPC and H-Bridge topologies. This issue will be later analyzed in detail employing the total active switching stress and utilization factors [30] as more precise comparison criteria.

### 2.1. Fundamental Principle

The 4L-QN topology provides four single-phase output voltage levels: $-v_d$, $-\frac{1}{2}v_d$, $\frac{1}{2}v_d$ and $v_d$. Each voltage level is generated by connecting the single-phase AC output terminal to the corresponding DC capacitor of the DC-link. The inverter configuration is based on six switching devices, divided in three pairs of switches that operate with complementary gating signals: $S_1 - \overline{S_1}$, $S_2 - \overline{S_2}$ and $S_3 - \overline{S_3}$. The previous working principle enables the use of classic pulse-width modulation (PWM) strategies, such as level-shifted (LS) and phase-shifted (PS) PWM, giving rise to $N = 23$ switching states which generate the four different output voltage levels as presented in Table 1.

**Table 1.** Allowed switching states and output voltages for the proposed four-level topology.

| # | $S_1$ | $S_2$ | $S_3$ | $v_{xN}$ |
|---|---|---|---|---|
| 1 | 0 | 0 | 0 | $-v_d$ |
| 2 | 0 | 0 | 1 | $-\frac{1}{2}v_d$ |
| 3 | 0 | 1 | 0 | $\frac{1}{2}v_d$ |
| 4 | 0 | 1 | 1 | $\frac{1}{2}v_d$ |
| 5 | 1 | 0 | 0 | $-v_d$ |
| 6 | 1 | 0 | 1 | $-\frac{1}{2}v_d$ |
| 7 | 1 | 1 | 0 | $v_d$ |
| 8 | 1 | 1 | 1 | $v_d$ |

According to the switching states presented in Table 1, the output voltage, can be expressed as in Equation (1)

$$v_{xN} = v_d\, S_1\, S_2 + \frac{1}{2}\, v_d\, \bar{S}_1\, S_2 + \frac{1}{2}\, v_d\, \bar{S}_2\, S_3 - v_d\, \bar{S}_2\, \bar{S}_3, \tag{1}$$

where $v_{xN}$ stands for the corresponding output voltage with respect to the neutral point $N$ and $v_d$ corresponds to the DC-link voltage; $S_k$ $k \in [1,\dots,3]$ stands for the corresponding active switch, $\bar{S}_k$ to its complementary switching state.

### 2.2. Modulation Scheme

PWM techniques for multilevel converters, such as selective harmonic elimination PWM (SHE-PWM), space vector PWM (SVPWM), and multi-carrier PWM (MCPWM) schemes have been extensively treated in literature [28,29,31]. Modulations techniques based on the use of multiple carriers, which constitute an extension of the classical two-level PWM methods, have become very popular because of its simplicity and effectiveness to control the output voltage in multilevel inverters. There exist various multi-carrier PWM techniques, which are distinguished by the type of carrier and modulating signals employed.

In this context, LS-PWM is suitable for its application in multilevel converter topologies because it ensures low harmonic distortion [32–34] and is simple on its implementation structure respect to space-vector methods. For the proposed inverter topology, a LS-PWM technique has been implemented, hence the output voltage is synthesized as follows [11]:

$$S_1 = \begin{cases} 1 & |u_{xn}^*| \geq |u_{c1}| \\ 0 & |u_{xn}^*| < |u_{c1}| \end{cases} \tag{2}$$

$$S_2 = \begin{cases} 1 & |u_{xn}^*| \geq |u_{c2}| \\ 0 & |u_{xn}^*| < |u_{c2}| \end{cases} \tag{3}$$

$$S_3 = \begin{cases} 1 & |u_{xn}^*| \geq |u_{c3}| \\ 0 & |u_{xn}^*| < |u_{c3}| \end{cases} \tag{4}$$

where $u_{xn}^*$ stands for the amplitude of the voltage reference corresponding to the $x$ phase, and $u_{c1}$, $u_{c2}$, $u_{c3}$ for the corresponding level-shifted carriers amplitude. The corresponding output voltage can be expressed as in Equation (5)

$$v_{xN} = v_d\, S_2\, S_3 \left[ S_1 + \frac{1}{2}\, \bar{S}_1 \right] - v_d\, \bar{S}_1\, \bar{S}_2 \left[ S_3 + \frac{1}{2}\, \bar{S}_3 \right]. \tag{5}$$

Implementation of the LS-PWM scheme enables only switching states 1, 2, 4 and 8 (as defined in Table 1) to be synthesized. Those corresponding switching states are shown in Figure 3

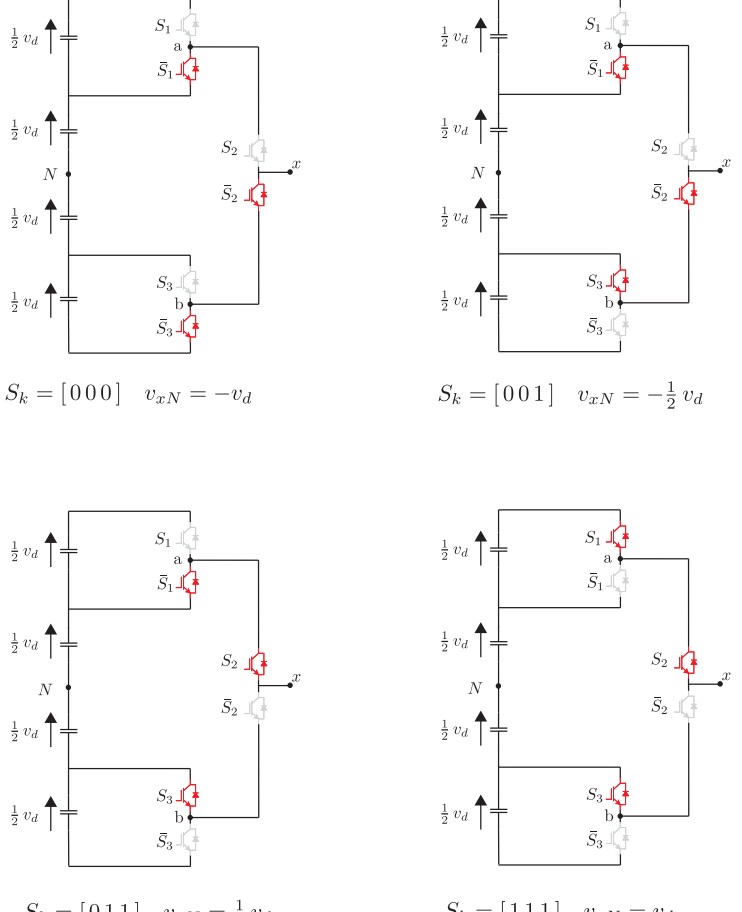

**Figure 3.** LS-PWM (level-shifted pulse-width modulation) scheme switching states.

In the LS-PWM scheme all the carrier signals are level shifted and in-phase with each other. The required number of carriers $N_c$ for a $n$ level converter is obtained by [11]:

$$N_c = n - 1. \tag{6}$$

For the 4L-QN topology, a 4-level LS-PWM has been implemented by using three in-phase carrier signals $C_{w\ell}$. Carrier signals $C_{w\ell}\ \forall\, \ell \in [1, \dots, n-1]$ are generated in the positive and negative axis, and properly scaled to obtain a range from $-1$ to $1$ pu (per unit).

In this way each carrier has an amplitude of $\frac{1}{3}$ pu. Figure 4 shows the implementation of the LS-PWM scheme for the proposed topology.

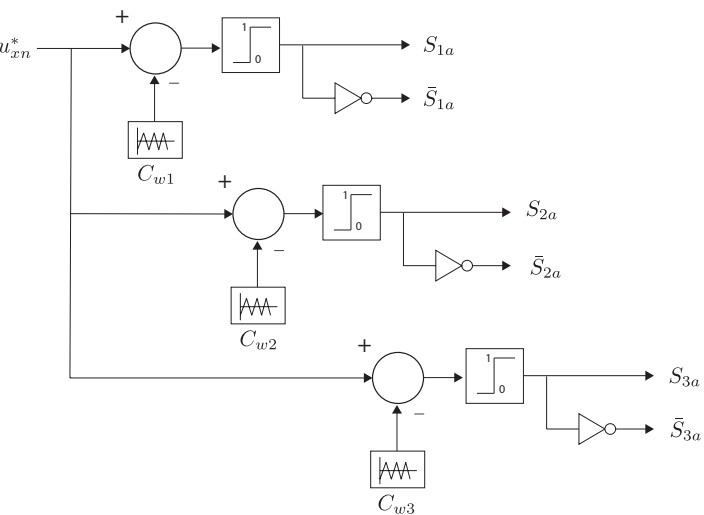

**Figure 4.** Implementation of the LS-PWM scheme.

Figure 5 shows the modulation pattern and the synthesized output voltage waveform using the previously described LS-PWM scheme. As presented in Figure 5b, the output voltage exhibits a four-level single-phase pattern.

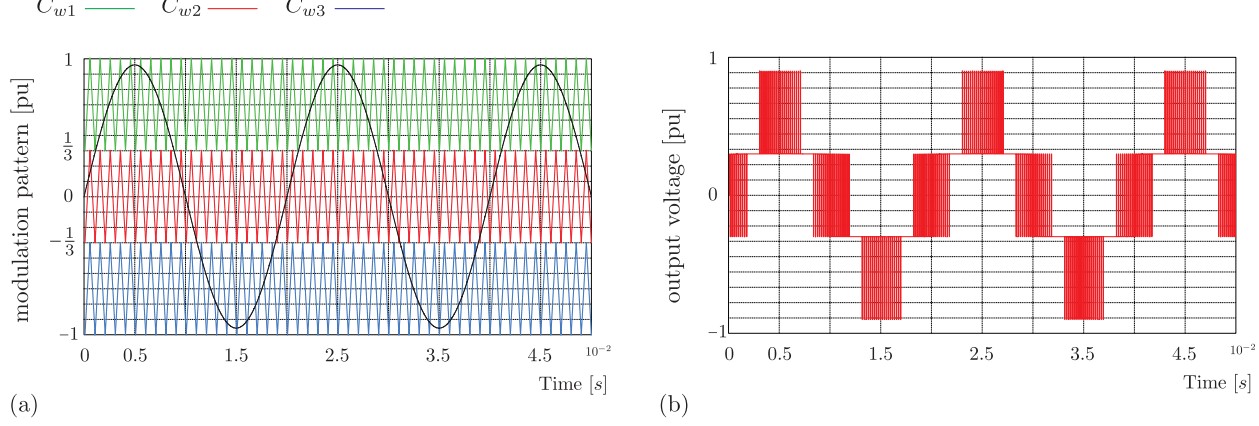

**Figure 5.** LS-PWM simulation results: (**a**) modulation pattern, (**b**) single-phase output voltage.

### 2.3. Balancing of DC-Link Capacitors

DC-link voltage unbalance is one of the main drawbacks in multilevel topologies. This effect is due to the asymmetries in the switching strategy, leading to nonuniform power distribution on each DC-link capacitor. In fact, in the particular case of LS-PWM not all capacitors are delivering energy on each duty cycle.

To deal with this problem a symmetrical distribution of the DC-link capacitors in each switching state is required, to compensate the asymmetrical charging and discharging process due to the modulation strategy. The particular DC-link configuration of the single-phase quasi-nested topology, ensures a symmetrical distribution of the DC-link capacitors. Moreover the absence of clamping devices, as in the nested configurations, ensures no common-mode circulating currents in the DC-link, thus reaching a natural voltage balance of the DC capacitors.

As shown in Figure 6, while $C_{1+}$, $C_{2+}$ are in the discharging region, $C_{1-}$, $C_{2-}$ are being charged and no common-mode circulating current is allowed to flow through $C_{1-}$, $C_{2-}$, thus ensuring an uniform voltage distribution within DC-link capacitors.

This fact constitutes a major advantage of this topology, when compared to the DSC three-phase converter, where the asymmetry of the DC-link with respect to the double-star configuration, and the absence of a fixed clamped neutral point, gives raise to a DC-voltage unbalance, thus requiring the implementation of a voltage drift compensation strategy as stated in [25,35].

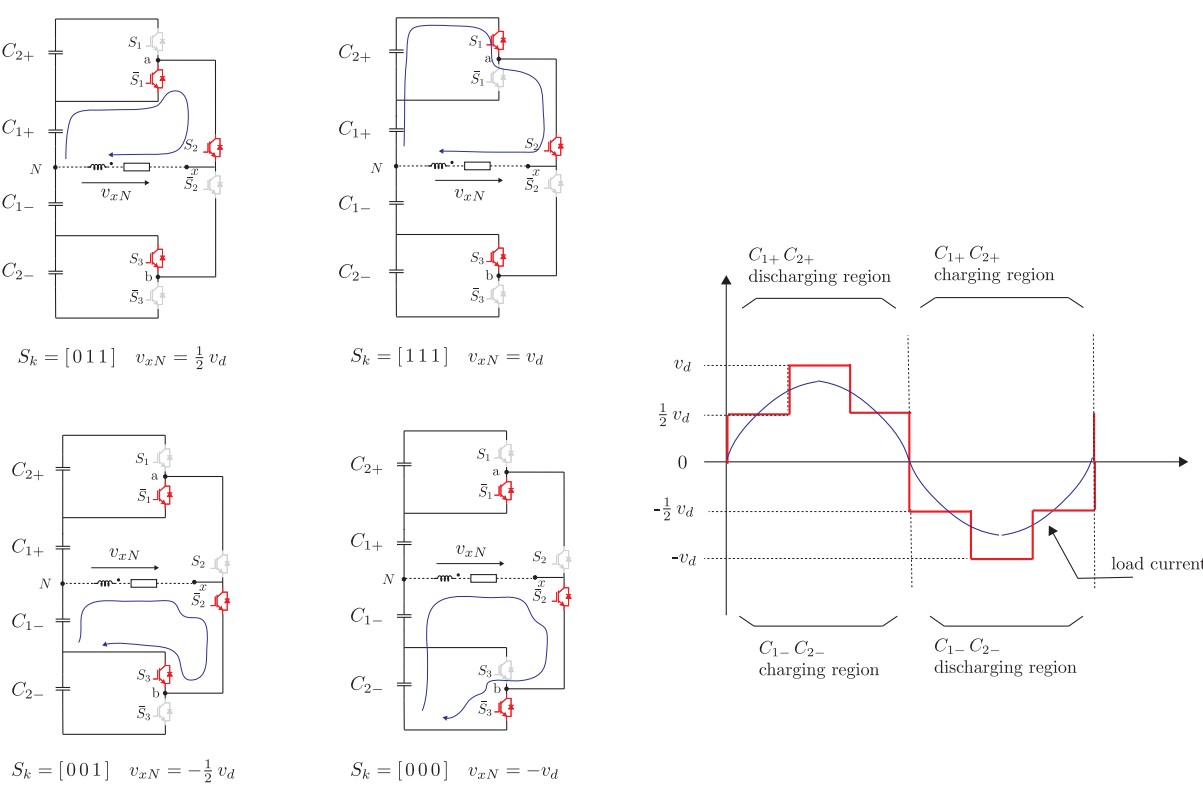

**Figure 6.** DC-link voltage balancing in the proposed topology.

## 3. Control Objectives

In this work a resonant current control scheme has been designed to ensure current regulation to a resistive-inductive single-phase load. The grid current reference $i_s^*$ is compared with the measured value of the single-phase load current $i_s$. The load current error signal is feed into a proportional multi-resonant (PMR) controller whose continuous transfer function is [36,37]:

$$C(s) = K_p + \frac{k_1 s}{s^2 + \omega_s^2} + \sum_{\ell=3,5} \frac{k_\ell s}{s^2 + \ell^2 \omega_\ell^2},\tag{7}$$

where $K_p$ is the proportional gain and $k_\ell$ is the resonant gain at each selected $\ell$-th harmonic. The implementation of the controller is shown in Figure 7.

The above controller design has been conceived in order to achieve selective harmonic impedance [38–40], by rejecting 3rd and 5th harmonic components. The controller resonant frequency $\omega_s$ is tuned to achieve resonance at the grid frequency, ensuring high gain and zero steady-state error.

The actuation of the PMR controller results in the voltage reference $u_{xN}^*$ which is feed into the LS-PWM modulator of the inverter. The block diagram of the load current control loop is shown in Figure 8.

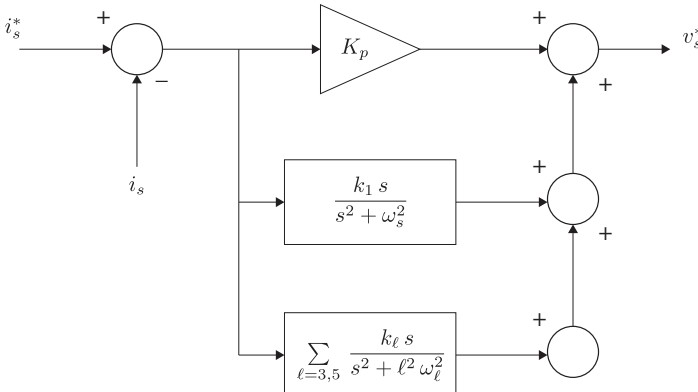

**Figure 7.** Proportional multi-resonant (PMR) load current control loop implementation.

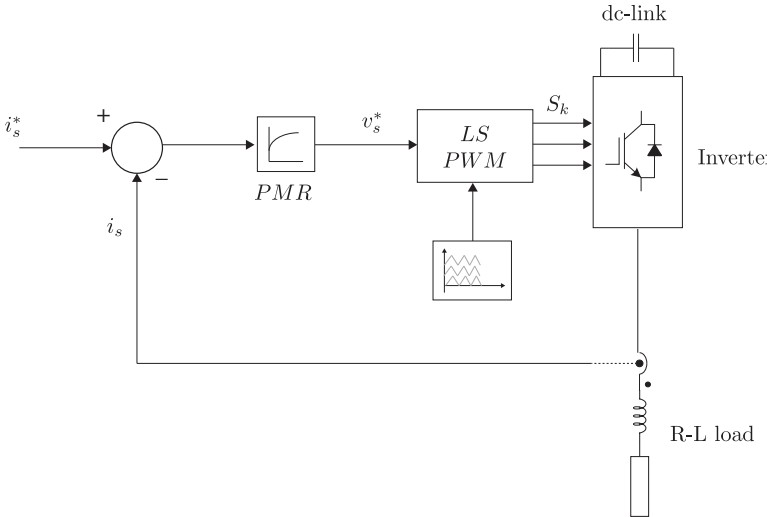

**Figure 8.** Block diagram of the single-phase current control loop.

In order to design the parameters of the controller, the PMR transfer function presented in (7) can be extended as follows:

$$C(s) = \frac{k_p\,s^2 + k_1\,s + k_p\,\omega_s^2}{s^2 + \omega_s^2} + \sum_{\ell=3,5} \frac{k_\ell\,s}{s^2 + \ell^2\,\omega_\ell^2}. \tag{8}$$

For the controller transfer function presented in (8), the primary design objective is related to the grid resonant frequency $\omega_s$ in order to ensure full reference tracking, so (8) may be decoupled, neglecting the $\ell^{th}$ filters, thus reaching the following transfer function:

$$C'(s) = \frac{k_p\,s^2 + k_1\,s + k_p\,\omega_s^2}{s^2 + \omega_s^2}, \tag{9}$$

where the zeros are placed using the pole-assignment method to accomplish with the desired controller bandwidth.

The design of the gains corresponding to each suppressed harmonic $k_\ell$ have effects only in the vicinity of their corresponding resonant frequency $\omega_\ell$, if the controller bandwidth is greater than each of the rejected frequencies $\omega_\ell$. Under this restriction $k_\ell$ can be obtained as follows [41,42]:

$$k_\ell = \frac{k_1}{\ell} \tag{10}$$

Discrete implementation of the PMR controller can be found by mapping the corresponding poles and zeros in (9) into the z-plane as $z = e^{s\,T_s}$ [43], with $T_s$ the sampling period, which yields:

$$C_d'(z) = k_p \frac{z^{-2} + \alpha_1\,z^{-1} + \alpha_0}{z^{-2} - 2\,\cos\left(\omega_s\,T_s\right)z^{-1} + 1}, \tag{11}$$

and the corresponding selective harmonic impedance transfer discrete function is given by:

$$C_d''(z) = \sum_{\ell=3,5} k_\ell \frac{z^{-2} - \cos\left(\omega_\ell\,T_s\right)z^{-1}}{z^{-2} - 2\,\cos\left(\omega_\ell\right)T_s\,z^{-1} - 1}. \tag{12}$$

## 4. Results

This section presents the simulation and experimental results of the proposed inverter topology. Simulation analysis has been performed using PLECS for modelling the control scheme, modulation stage, inverter converter topology, and load. The analysis was completed by using the same scenarios of the experimental set-up in order to improve the conceptual verification. The key simulation and experimental parameters are identified in Table 2. It is important to highlight that experimental validation parameters have been selected according to the reduced power experimental prototype.

**Table 2.** Simulation and experimental parameters.

| Symbol | Parameter | Simulation Value | Experimental Value |
|--------|-----------|------------------|--------------------|
| **Inverter Parameters** | | | |
| $v_d$ | Total DC-link voltage | 540 [V] | 90 [V] |
| $C_{DC}$ | DC-link capacitors | 2400 [µF] | 2400 [µF] |
| $f_{cr}$ | Carrier frequency | 5000 [Hz] | 5000 [Hz] |
| $L_s$ | Load inductance | 10 [mH] | 10 [mH] |
| $R_s$ | Load resistance | 10 [Ω] | 10 [Ω] |
| **Control Parameters** | | | |
| $T_s$ | Sample period | 15 [µs] | 15 [µs] |
| BW | Current control bandwidth | 300 [Hz] | 300 [Hz] |

### 4.1. Simulation Results

Simulation results for three different operational conditions have been performed, for different current references, which were selected in order to show the performance of the output voltage and current when operating within the linear modulation index $m$ boundaries $0 < m < 1$ and also in the over-modulation zone $m > 1$.

Figure 9 shows the output voltage and current waveforms for three different current references; in Figure 9a,b the resulting modulation index is within the linear modulation boundaries resulting in a symmetrical distribution of the switching states, hence a symmetrical energy distribution of the DC sources; in Figure 9c the resulting modulation index exists in the over-modulation region presenting a saturation effect, which results in the lack of transferred energy (charging and discharging) of the DC capacitors. This asymmetry in the DC capacitors voltage distribution results in a distortion of the load current waveform, as shown in Figure 9c where the controller is unable to track the desired current reference. These facts constitute the fundamental control limitation of the load current.

The transient performance of the PMR current controller is shown inf Figure 10. A reference step change is introduced at time $t = 0.5\,\mathrm{s}$ from 0 to 1 pu. Later at $t = 1\,\mathrm{s}$, while the controller is tracking a 1 pu current reference, a load impact of 100% is applied. As can be observed in the results, the controller has a fast transient response with no observable steady-state error.

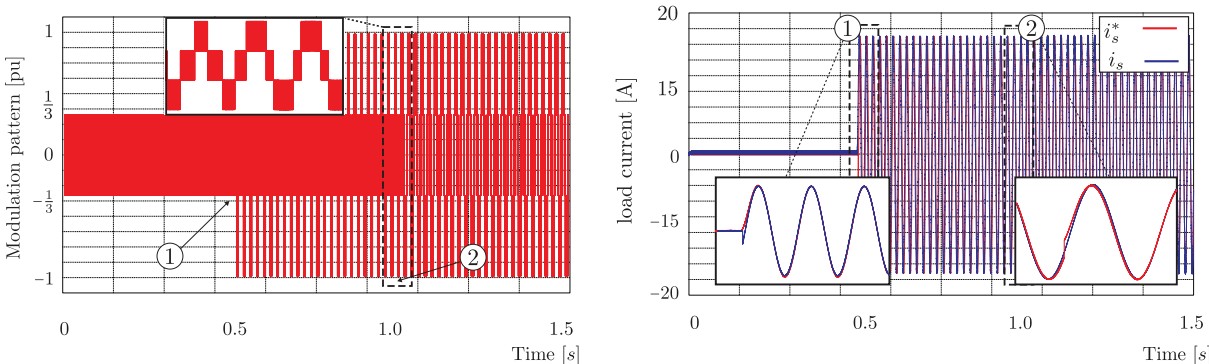

**Figure 9.** Simulation results. Inverter voltage and load current with modulation index in the linear range $m = 0.5$ (**a**), $m = 0.8$ (**b**) and in the over-modulation region $m = 1.5$ (**c**).

**Figure 10.** Simulation results: transient performance of the PMR current controller. (1) Reference step change from 0 to 1 pu at $t = 0.5$ s. (2) Load impact of 100% at $t = 1$ s.

### 4.2. Voltage and Current Stress of the Semiconductor Switches

The assessment of voltage and current stress of the active switches in the 4L-QN inverter has been carried out by simulation with PLECS software. The transient characteristics of the specific IGBTs employed to build the experimental set-up (Semikron SKM75GB12V) were included in the simulation using the data provided by the manufacturer.

Figure 11 shows the voltage and current waveforms in the active switches of the proposed inverter. The peak blocking voltages for the active switches are $\frac{1}{4}$ and $\frac{3}{4}$ of the DC-voltage, whereas the peak current in any of the switches remains lower than $0.05\,\text{pu}$, normalized with respect to the nominal current rating of the device (75 A).

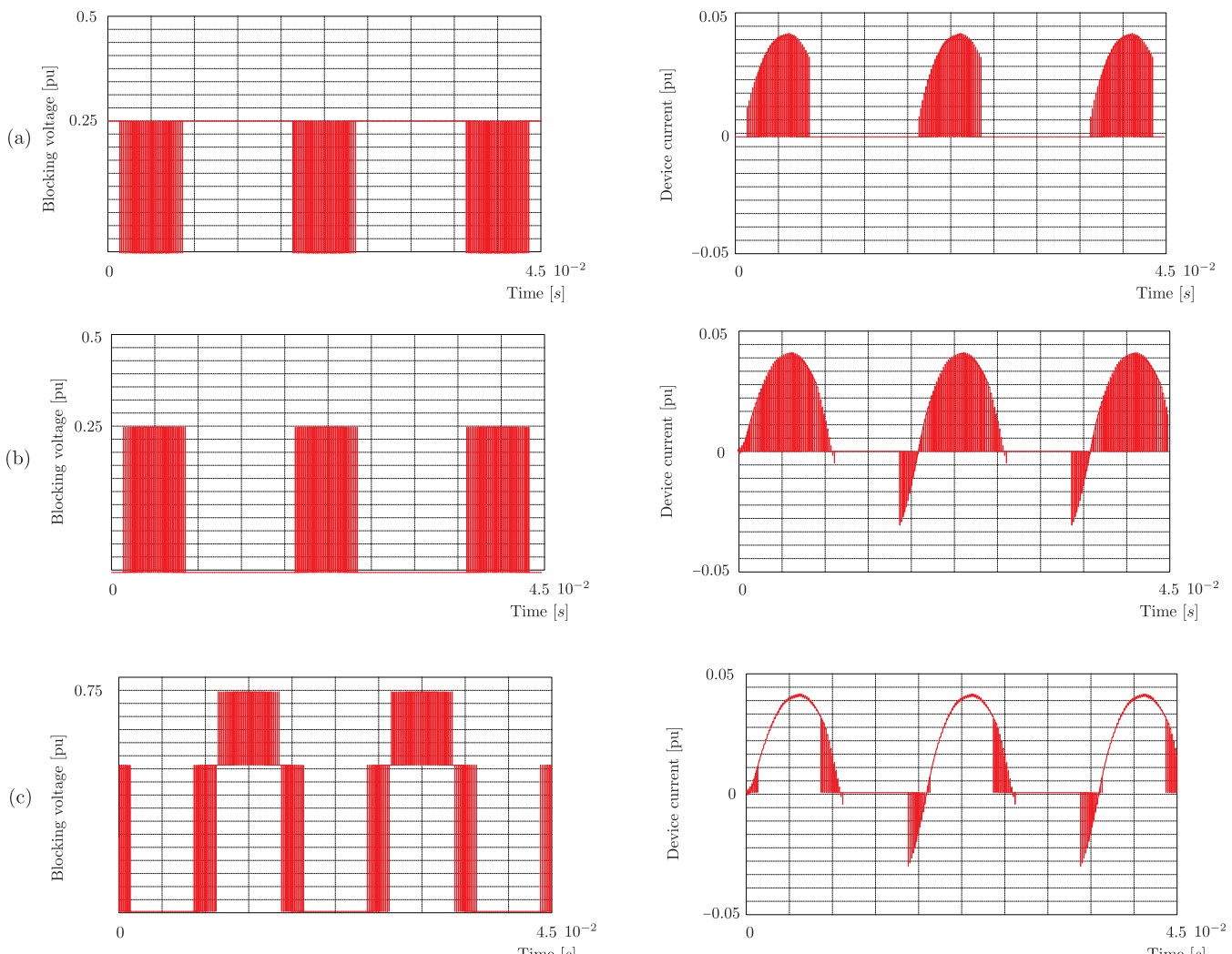

**Figure 11.** Voltage and current waveforms of the semiconductor switches in the four-level quasi-nested (4L-QN) inverter. (**a**) Switches $S_1, S_3$; (**b**) switches $\overline{S_1}, \overline{S_3}$; (**c**) switches $S_2, \overline{S_2}$.

A quantitative measure of the switch stress is defined by the active switching stress and utilization factors [30]. The total active switch stress in a converter with $k$ switches is defined as:

$$S = \sum_{j=1}^{k} V_j I_j, \tag{13}$$

where $V_j$ and $I_j$ are the peak voltage and current applied to the $j-$th switch, respectively. Lower values of $S$ are preferable due to the reduction of semiconductor costs. On the other hand, for a given load active power $P$, the active switch utilization is defined as:

$$U = \frac{P}{S}. \tag{14}$$

Higher utilization factors are related to better cost-efficiency of the active switches in the converter.

Using simulation results, the performance indices defined by (13) and (14) were applied to the 4L-QN inverter and compared with those of the NPC, ANPC and CHB. A power load $P = 360\,\text{W}$ was considered, which is equivalent with the rating of the experimental setup. The results are presented in Table 3. Among the four topologies compared in the table, the proposed 4L-QN inverter achieved the second lowest active stress factor, after NPC. The low score of NPC in this evaluation is because the clamping diodes are not included in the stress calculation. Similarly, the proposed inverter showed a good performance in the comparison, with a higher utilization factor than the H-bridge and the ANPC inverters.

**Table 3.** Switch stress and utilization.

| Topology | Active Switch Stress ($S$) | Utilization Factor ($U$) |
|---|---|---|
| H-Bridge | 1915.2 | 0.188 |
| NPC | 1103.4 | 0.326 |
| ANPC | 1825.7 | 0.197 |
| 4L-QN | 1729.4 | 0.208 |

*4.3. Efficiency Analysis*

The efficiency of the proposed 4L-QN inverter has been studied and compared with conventional topologies by using simulation analysis with PLECS software. The switching characteristics of the Semikron SKM75GB12V IGBT used in the experimental setup were considered for the analysis. Table 4 contain the results of the efficiency analysis for the different topologies used for comparison: H-Bridge, NPC, ANPC and the proposed 4L-QN inverter. From the results presented in the Table, the proposed inverter offers conduction losses 6.25% lower than ANPC, 24.08% lower than NPC and 31.15% lower than H-bridge topologies. The switching losses in the proposed inverter are 47.37% lower than ANPC, 90.24% lower than NPC and 83.33% lower than H-bridge configurations. The efficiency tests have been performed using a 5 kHz switching frequency and a modulation index $m = 0.8$ for all the evaluated topologies.

**Table 4.** Semiconductor switches losses.

| Topology | Conduction Losses (W) | Switching Losses (mJ) |
|---|---|---|
| H-Bridge | 8.18 | 12 |
| NPC | 7.31 | 20.5 |
| ANPC | 5.92 | 3.8 |
| 4L-QN | 5.55 | 2 |

*4.4. Experimental Validation*

The experimental test rig comprises three independent isolated DC voltage sources, using the middle point terminal of one of the DC sources to serve as neutral point connection. This arrangement was connected to the DC-link capacitors as shown previously in Figure 2. A dSPACE 1103 digital control platform was used to run the control algorithm and modulation strategy. Complementary switching logic and dead-time generation, were hardware-implemented within the designed gate drive PCBs. Figure 12 shows part of the experimental quasi-nested inverter, control platform and the single-phase load.

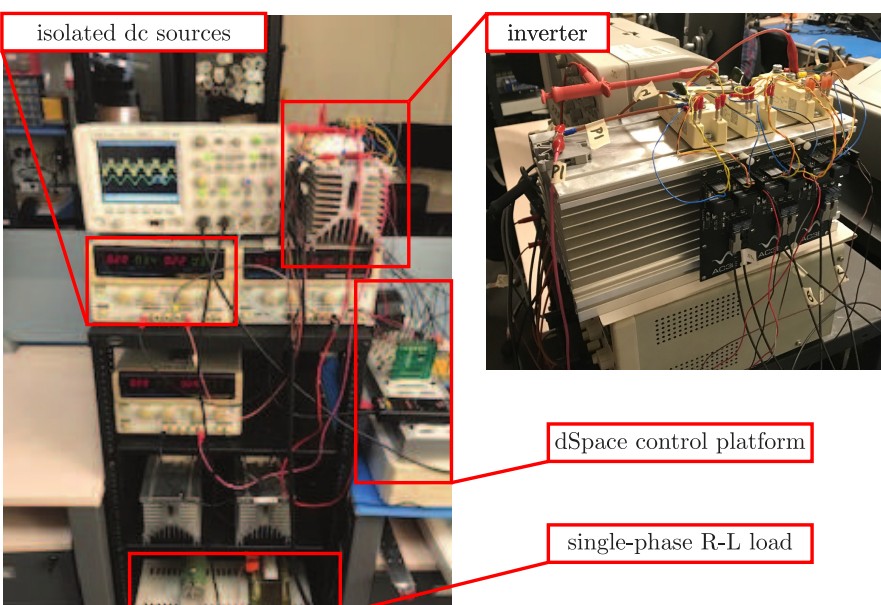

**Figure 12.** Experimental setup of the 4L-QN inverter.

Experimental validation considered three different current references, selected in order to show the inverter performance when operating within linear modulation region and in over-modulation zone. In Figure 13a,b, output voltage and current waveforms are shown for a reference modulation index within linear modulation boundaries.

The PMR controller was designed using the pole placement method for the desired bandwidth and implemented in C code, within the dSPACE 1103 platform algorithm. The chosen controller bandwidth was according to the processing capability of the control platform.

As shown in Figure 13a,b, within linear modulation region, the voltage output waveform exhibits symmetrical switching distribution resulting in a sinusoidal load current.

Figure 13c shows the experimental results for the output voltage and load current, when operating in the over-modulation region. As shown in the obtained results, a non-symmetrical switching voltage distribution builds-up causing the load current to present a distorted sinusoidal waveform.

*4.5. Discussion*

The proposed quasi-nested topology exhibits a good performance in terms of symmetry of the output voltage waveforms and active switching stress with respect to classical multilevel topologies. It is a very attractive topology for single-phase applications where multiple isolated DC sources, are required, such as PV with multiple strings configurations. It is also can be adopted as solution for single DC source and DC-link capacitors configurations.

Proportional multi-resonant control has proved to be a good alternative to traditional PI load current control schemes for single-phase applications, avoiding the extra requirement of phase locked-loop (PLL) synchronization and complex single-phase decoupled vector control, which requires an orthogonal estimated dynamic system [40].

Due to laboratory equipment and test bench design implementation limitations, the proposed topology was validated, feeding a linear resistive-inductive load. Despite of these limitations, an experimental validation has been performed, fulfilling the requirements in terms of output voltage levels and control of the load current, without the need of a PLL, ensuring satisfactory reference tracking within the linear modulation region.

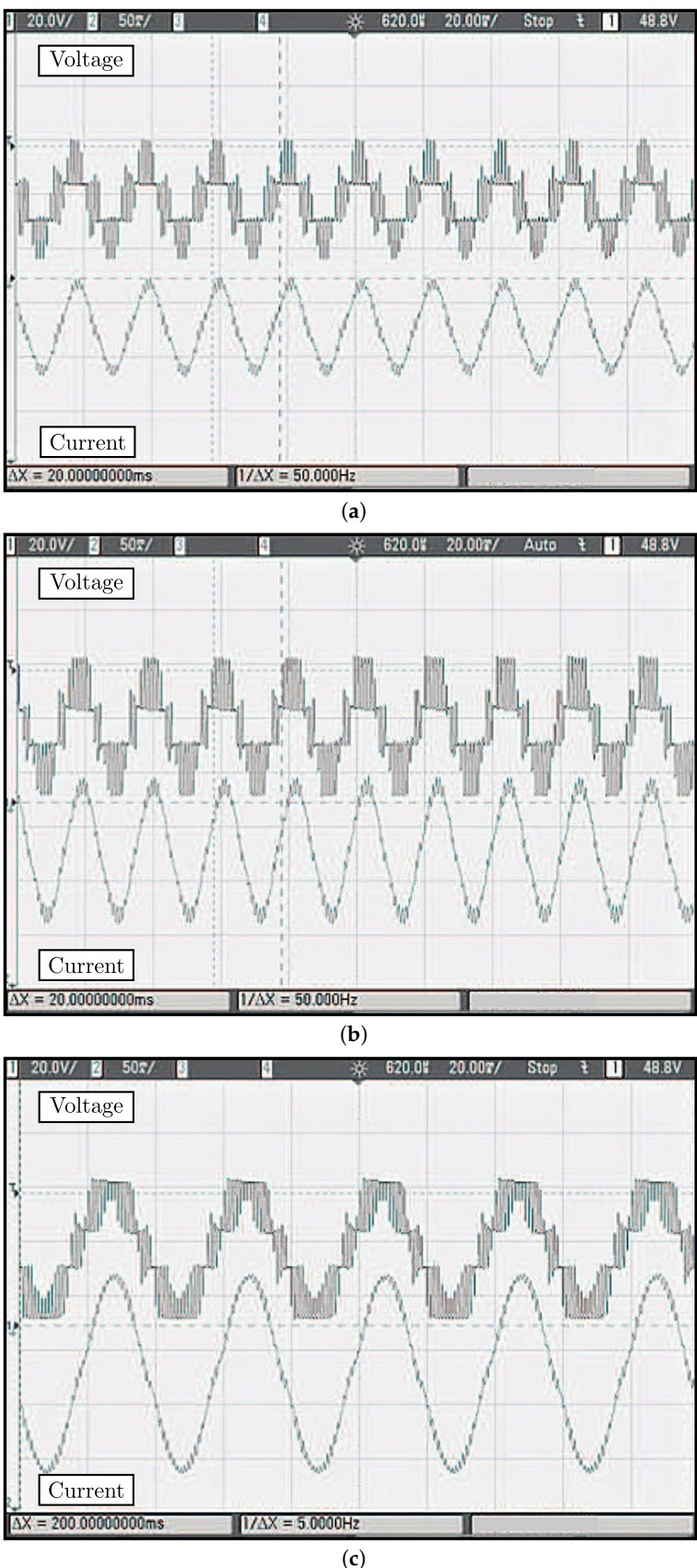

**Figure 13.** Experimental results for voltage and current waveforms with modulation index (**a**) *m* = 0.5, (**b**) *m* = 0.8 and (**c**) *m* = 1.5.

Figure 14 presents a comparison with classical topologies, for single-phase applications, in terms of their number of levels, blocking voltage, THD$_v$, advantages and disadvantages. The proposed 4L-QN topology exhibits many benefits in terms of simplicity of the modulation strategy, low active switch stress and higher number of voltage levels. The 4L-QN topology achieves a better performance in terms of the voltage THD than conventional MLIs, which comes from the additional voltage level that the proposed inverter is able to generate in comparison to other topologies.

| Topology | 4L-QN | 3L-NPC | 3L-ANPC | H-Bridge |
|---|---|---|---|---|
| Power circuit (fundamental cell) |  |  |  |  |
| # of levels | 4 | 3 | 3 | 3 |
| # of semi-conductors | 6 active | 4 active + 2 diodes | 6 active | 4 active |
| Blocking voltage | $\frac{1}{4}v_d$ ; $\frac{3}{4}v_d$ | $\frac{1}{2}v_d$ | $\frac{1}{2}v_d$ | $\frac{1}{2}v_d$ |
| Modulation | LS-PWM | LS-PWM + balance | LS-PWM + Clamp.Sw. + balance | SPWM |
| THD$_v$ |  |  |  |  |
| Advantages | No clamping devices required. Lower active switch voltage stress compared to other multilevel topologies. Natural balance of DC-capacitors. Lower losses. Smaller dc-capacitors | Mature topology. Modulation using classical multi-carrier and space-vector based PWM techniques. | Clamping switches ensure equal switching frequency. Handles 32% more power compared to NPC topology. | Mature topology. Modulation using single carrier SPWM techniques. Single dc-capacitor required. |
| Disadvantages | Requires more dc-link capacitors compared to classical multilevel topologies. | Neutral clamping diodes. Dc-voltage unbalance requires balancing strategy. | Neutral clamping diodes. Dc-voltage unbalance requires balancing strategy. | Larger capacitor required compared to multilevel topologies. Higher THD$_v$ compared with multilevel topologies. |
| References | (this topology) | [4], [5] | [4], [6] | [8],[9] |

**Figure 14.** Comparison of the proposed topology and conventional single-phase multilevel converters.

Figure 15 shows the output voltage spectrum of the proposed 4L-QN inverter operating with LS-PWM and a modulation index $m = 0.8$, and a comparison with the voltage spectrum of H-Bridge, NPC and ANPC inverters. All the inverter topologies generate a voltage spectrum with low harmonic content at low frequencies, and the characteristic PWM switching harmonics distribution around a switching frequency of 5 kHz. The proposed 4L-QN inverter yields the lowest THD$_v$ among the evaluated topologies, with 65% less than H-Bridge and 32.69% less than NPC and ANPC.

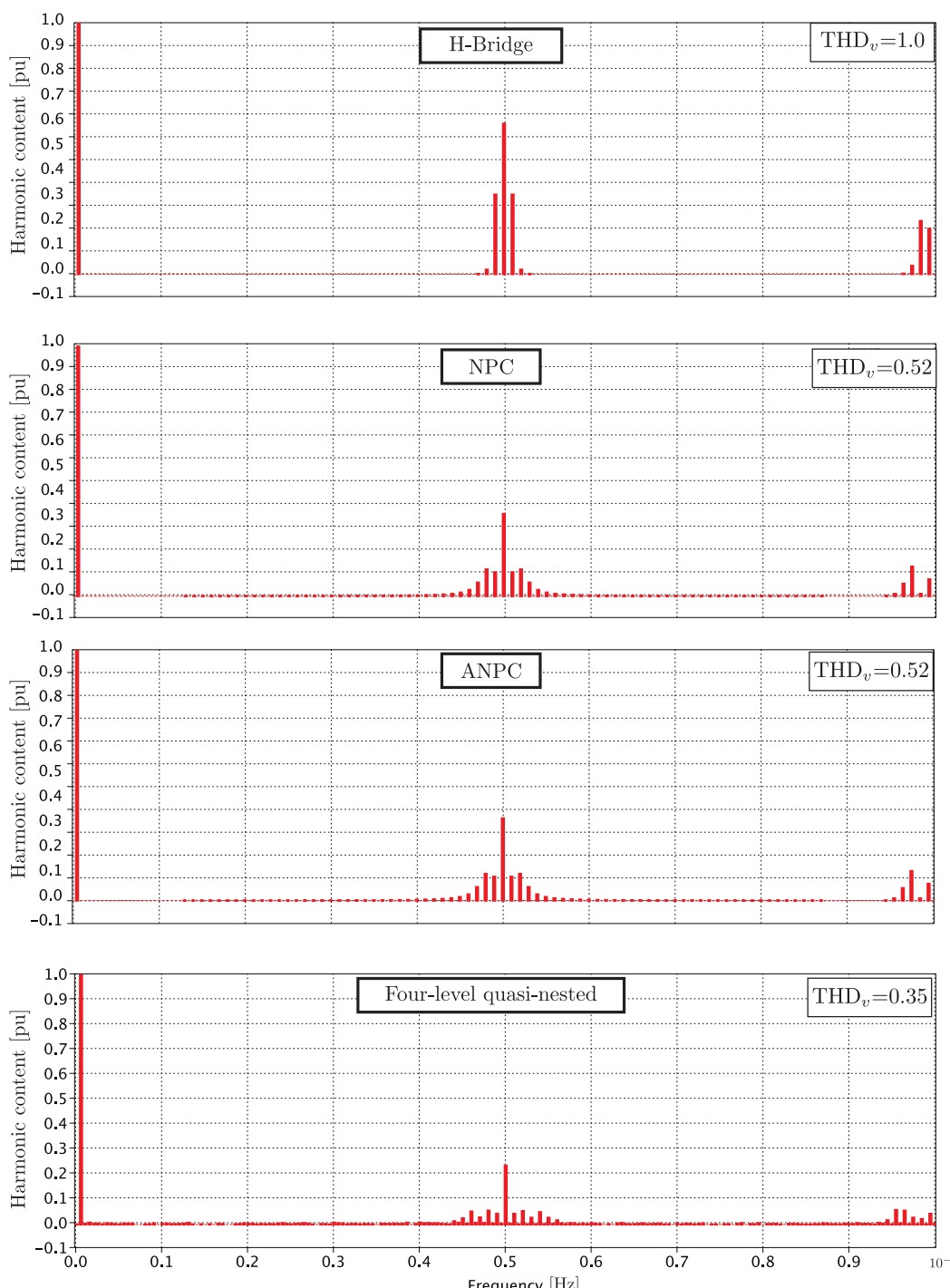

**Figure 15.** Simulation result: voltage harmonic spectrum of the 4L-QN inverter and conventional counterparts ($m = 0.8$).

## 5. Conclusions

In this work a novel multilevel topology based on a quasi-nested structure, for its application in single-phase systems was described and validated via simulation analysis and experimental results.

The proposed topology, modulation and control scheme were validated experimentally in a reduced scale power prototype with a straightforward implementation. Additionally, a load current control strategy based on the use of a multi-resonant proportional controller was evaluated, ensuring full reference tracking with zero steady-state error, for sinusoidal references.

The proposed inverter topology merged the benefits of multilevel inverters with the use of isolated DC sources for single-phase applications. Moreover, despite NPC and ANPC multilevel topologies, DC-link capacitors do not require a balancing strategy, ensuring DC-link capacitors self balancing.

Future research on the proposed MLI will be focused on the development of new control strategies specifically designed for this topology, reactive power control for grid-connected applications and interleaved configuration for enhanced harmonic performance.

**Author Contributions:** Conceptualization, C.A.R. and H.Y.; methodology, C.A.R.; software, C.A.R.; validation, C.A.R.; formal analysis, C.A.R. and H.Y.; investigation, C.A.R. and H.Y.; writing—original draft preparation, C.A.R. and H.Y.; writing—review and editing, C.A.R. and H.Y. All authors have read and agreed to the published version of the manuscript.

**Funding:** This research received financial support from the Chilean Found for Human Resource Development (ANID) through its Ph.D. scholarships (CONICYT/21130448) and from the Research Direction of the Universidad de La Frontera.

**Institutional Review Board Statement:** Not applicable.

**Informed Consent Statement:** Not applicable.

**Data Availability Statement:** Data is contained within the article.

**Acknowledgments:** The authors wish to thank the support of the Electrical Engineering School, Pontificia Universidad Catolica de Valparaiso, and also the support provided by the Advanced Center for Electrical and Electronic Engineering AC3E (ANID/FB0008) of Universidad Tecnica Federico Santa Maria.

**Conflicts of Interest:** The authors declare no conflict of interest.

## Abbreviations

The following abbreviations are used in this manuscript:

| | |
|---|---|
| 3L-ANPC | Three Level Active Neutral Point Clamped |
| 3L-FC | Three Level Flying Capacitor |
| 3L-NPC | Three Level Neutral Point Clamped |
| 4L-FC | Four Level Flying Capacitor |
| 4L-QN | Four-level quasi-nested |
| AC | Alternating Current |
| DC | Direct Current |
| ANPC | Active Neutral Point Clamped |
| CHB | Cascaded H-Bridge |
| DSC | Double-star converter |
| FC | Flying Capacitor |
| IGBT | Isolated Gate Bipolar Transistor |
| LS-PWM | Level-Shifted Pulse Width Modulation |
| MOSFET | Metal Oxide Semiconductor Field Effect Transistor |
| MLI | Multi-level Inverter. |
| MV | Medium-Voltage |
| NLC | Nearest level control |
| NPC | Neutral Point Clamped |
| PI | Proportional Integral |
| PLL | Phase Locked Loop |
| PMR | Proportional Multi Resonant |
| PS-PWM | Phase-Shifted Pulse Width Modulation |
| PWM | Pulse Width Modulation |
| R-L | Resistive-inductive |
| SHE-PWM | Selective harmonic elimination PWM |
| SVPWM | Space Vector PWM |
| THD | Total Harmonic Distortion |

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
