# Peer review of "Four-Level Quasi-Nested Inverter Topology for Single-Phase Applications"

_electronics, doi:10.3390/electronics10030233_

Round 1

Reviewer 1 Report

In this paper, a single-phase MLI topology derived from the 4-level double-star converter is developed. The converter topology, operating principles, switching states and modulation scheme are analyzed. Simulation results and experimental validation are given. The idea is interesting. In my opinion, the author should address the following issues:

1)The authors should further explain the advantages and disadvantages of the proposed topology compared to other multilevel topologies. Although the table is given, quantitative comparisons are recommended.

2)In the experiment, why is the DC-side voltage set so low? A clearer figure of the experimental platform, especially for the inverter, is recommended.

3)The simulation and experiment results should be improved.

Author Response

Reviewer’s comment: In this paper, a single-phase MLI topology derived from the 4-level double-star converter is developed. The converter topology, operating principles, switching states and modulation scheme are analyzed. Simulation results and experimental validation are given. The idea is interesting. In my opinion, the author should address the following issues:

Author response: Thank you very much for your constructive comments. We humbly hope the revised paper contains acceptable responses to the issues addressed by the reviewer.

Comment #1: The authors should further explain the advantages and disadvantages of the proposed topology compared to other multilevel topologies. Although the table is given, quanti- tative comparisons are recommended.

Author response: We thank the reviewer for this helpful suggestion.

Author action: The revised version of the manuscript now contains additional quantitative indices for comparison of the proposed topology. Section 4.2 now contains an active switch stress analysis, an efficiency analysis was included in section 4.3 and a comparison of THD and harmonic spectra has been carried out in section 4.5.

Comment #2: In the experiment, why is the DC-side voltage set so low? A clearer figure of the experimental platform, especially for the inverter, is recommended.

Author response: In the experimental result a DC voltage of 90V was used. This reduced voltage level was selected according to the physical limitations of our laboratory equipment, and due to safety considerations.

Author action: We have modified Fig. 9 in the revised manuscript, to include a more detailed view of the inverter experimental setup.

Comment #3: The simulation and experiment results should be improved.

Author response: Thank you for your valuable suggestion. New simulation results have been ob- tained and included in the revised version of the paper. Unfortunately, we were not able to perform new experimental trials due to access restrictions to our Institution’s labs. This is because the national measures imposed in response to the Covid-19 outbreak. However, we humbly hope that the new si- mulation results and additional analysis included in the revised paper meet with the reviewer’s approval.

Author action: New simulation results have been included in section 4.2 in order to analyze the transient performance of the current controller implemented with the proposed four-level inverter. Vol- tage and current waveforms in the semiconductor switches of the proposed topology were also included. Moreover, section 4.5 of the revised manuscript contains the harmonic spectrum of voltage obtained by simulation of the proposed inverter and conventional multi-level topologies used for comparison purposes.

Reviewer 2 Report

The paper proposes a four-level single-phase inverter.

The paper is well written and well organized. However, the following are some questions and recommendations that the authors should address to improve the paper:

  1. Please provide waveforms for a transition from no load to full load and waveform of m=0 output voltage to m=0.8 output voltage mode.
  2. In the grid-tied performance, Does this converter support non-unity power factors (reactive power)? If yes, please explain the operation and provide related waveforms.
  3. Efficiency analysis should be properly presented in this paper.
  4. The voltage and current stress of the proposed converter should be shown and compared with their counterparts.
  5. The Authors should provide the comparison of Utilization factor as it was described in “Fundamentals of Power Electronics SECOND EDITION by Robert W. Erickson, Dragan Maksimovic, section 6.4”
  6. The harmonic spectrum of output voltage for the proposed converter and converter in Fig. 13 should be provided.
  7. The controller performance should be tested experimentally against source and load disturbances.
  8. The waveforms of the voltage of switches should be provided.
  9. Recently, the single-phase inverter idea has been shown helpful to fulfil the current and the voltage regulation. Some discussions on single-phase inverter should be provided, and these recently published results will provide some research ideas, e.g., "High-buck in Buck and High-boost in Boost Dual-Mode Inverter", "Single-Phase Dual-Mode Interleaved Multilevel Inverter for PV Applications", "An Integrated Interleaved Dual-Mode Time-Sharing Inverter for Single Phase Grid Tied Applications"

Author Response

Reviewer’s comment: The paper proposes a four-level single-phase inverter. The paper is well written and well organized. However, the following are some questions and recommendations that the authors should address to improve the paper:

Author response: We are sincerely grateful for your detailed and professional review of our work. We have thoroughly studied your comments and hope the revised version of our manuscript will meet with your approval.

Comment #1: Please provide waveforms for a transition from no load to full load and waveform of m=0 output voltage to m=0.8 output voltage mode.

Author response: Thanks for your pertinent request. The operating conditions described by the re- viewer were created using the closed-loop current control implemented for the proposed inverter. This allowed us to obtain waveforms that show fast transitions from low to high modulation index, and the reaction to a load step change.

Author action: Waveforms showing the performance of the system against load and reference step changes have been added to the revised manuscript in section 4.1.

Comment #2: In the grid-tied performance, does this converter support non-unity power factors (reactive power)? If yes, please explain the operation and provide related waveforms.

Author response: The proposed topology is in deed capable of reactive power control, as demonstrated in a three-phase application for grid-connected PV systems [R1]. In this paper we have implemented a single-phase MLI with a proportional multi-resonant controller, which simplifies the current control by avoiding the need of PLL synchronization and single-phase orthogonal decomposition. Therefore, tes- ting the reactive power control would require the implementation of a suitable decoupled power control scheme which is currently out of the scope of this work. However, we acknowledge the relevance of this topic and will consider its proper development in the future.

[R1] Rivera, S., Kouro, S., Llor, A., & Reusser, C. (2017). Four-level double star multilevel conver- ter for grid-connected photovoltaic systems. 2017 19th European Conference on Power Electronics and Applications, EPE 2017 ECCE Europe, 2017-January. https://doi.org/10.23919/EPE17ECCEEurope.2017.8099405

Comment #3: Efficiency analysis should be properly presented in this paper.

Author response: Thank you for your valuable suggestion.

Author action: We have included an efficiency analysis based on simulation with the specialized soft- ware PLECS, employing accurate manufacturer data of the same IGBT used in the construction of the experimental inverter (Semikron SKM75GB12V). The results and discussion of the efficiency analysis are included in section 4.3 of the revised manuscript.

Comment #4: The voltage and current stress of the proposed converter should be shown and compared with their counterparts.

Author response: We agree with the reviewer on the importance of voltage and current stress analysis of the proposed topology, and the comparison with its counterparts.

Author action: Considering your recommendation, in section 4.2 of the revised manuscript we have included simulated waveforms for blocking voltage and current of the switches of the proposed topology. The comparison of the active switch stress has been addressed using the performance indices suggested in your next comment. A revision on the comparison of the blocking voltage of active switches is pre- sented in section 4.5, where the comparative table has been modified according to the new results.

Comment #5: The Authors should provide the comparison of Utilization factor as it was described in Fundamentals of Power Electronics SECOND EDITION by Robert W. Erickson, Dragan Maksimovic, section 6.4.

Author response: Many thanks for your insightful suggestion.

Author action: We have included the performance indices recommended by the reviewer for the analy- sis and comparison of the proposed inverter with other topologies. We have employed detailed simulation results using the specialized software PLECS, including the transient response of the semiconductors implemented in the experimental setup (Semikron SKM75GB12V IGBT). The results are presented and discussed in section 4.2 of the revised paper.

Comment #6: The harmonic spectrum of output voltage for the proposed con- verter and converter in Fig. 13 should be provided.

Author response: We appreciate your pertinent suggestion.

Author action: The output voltage harmonic spectrum of the proposed inverter and its conventional counterparts have been added in section 4.5 of the revised manuscript. The harmonic distribution and THD obtained with the four configurations are analyzed and compared.

Comment #7: The controller performance should be tested experimentally against source and load disturbances.

Author response: Thank you for your pertinent suggestion. Unfortunately, currently we do not have access to our institution’s laboratory facilities due to the Covid-19 outbreak. This makes impossible to obtain additional experimental results, as the one requested by the reviewer. As a substitution, we have considered simulation results using the specialized PLECS software with accurate mathematical models of the inverter, load and control system. We sincerely hope the reviewer will understand our situation and find this approach acceptable.

Author action: A figure presenting simulation results of dynamic performance of the proportional multi-resonant (PMR) current controller has been added in section 4.1 of the revised manuscript.

Comment #9: Recently, the single-phase inverter idea has been shown helpful to fulfil the current and the voltage regulation. Some discussions on single-phase inverter should be provided, and these recently published results will provide some research ideas, e.g., High-buck in Buck and High-boost in Boost Dual-Mode Inverter, Single-Phase Dual-Mode Interleaved Multilevel Inverter for PV Applications, An Integrated Interleaved Dual-Mode Time-Sharing Inverter for Single Phase Grid Tied Applications.

Author response: We are very thankful to you for bringing to our knowledge these interesting and recent applications of single-phase inverters.

Author action: We have cited all the recommended references in the Introduction of the revised ma- nuscript, in order to highlight the relevance of single-phase inverter applications.

Reviewer 3 Report

Have you considered why using four levels in this study? Justification of the number of levels is required before setting up the motivation/contribution of this work. Without this clarification, this work only looks like a lab report, with significant gaps towards a research paper.

Both the abstract and introduction are a significant lack of quantitative evidence. Only qualitative descriptions are provided without any numbers to support the statement, which is neither informative nor convincing.

The key experimental performance of the proposed inverter is missing in the abstract.

Introduction: the authors made a very high-level comparison statement of different MLI technologies among cascaded H-Bridge (CHB), the neutral-point clamped (NPC), and the flying capacitor (FC). First, the schematic of each topology is suggested to add for illustration. Second, a more critical assessment of their disadvantages is required, by supplying some numbers. For instance, "NPC its excessive component count limits the
expansion of the system to higher number of levels, requiring also complex capacitor voltage-balancing
schemes." Please consider answering these questions:
How excessive component count? How many components for achieving higher number of levels? Why we need a higher number of levels? What is the optimum number of levels?
Similar questions to clarify for FC: "FC has the drawback of requiring a high number of large capacitors" - How many capacitors and how large are they?

There is a lack of up-to-date literature survey for the modular multilevel converter/inverter part and the modulation strategy - some references are outdated (5-10 years ago). The following paper on the state of the art is recommended to cite.
J. Li, "Design and Control Optimisation of a Novel Bypass-embedded Multilevel Multicell Inverter for Hybrid Electric
Vehicle Drives," 2020 IEEE 11th International Symposium on Power Electronics for Distributed Generation Systems
(PEDG), Dubrovnik, Croatia, 2020, pp. 382-385, doi: 10.1109/PEDG48541.2020.9244313.

The authors presented Figure 1 in the Introduction without mentioning it. It should be placed in section 2.
Figures 10-12: Are these waveform images taken from experimental photos or screenshots of the instrument? They are with extremely low resolution (cannot read it without zooming in) and without notation of which is voltage, and which is current. Suggest to save the data, process them, and replot.

dc should be capitalized throughout the manuscript.

Author Response

Comment #1: Have you considered why using four levels in this study? Justi- fication of the number of levels is required before setting up the motivation/contribution of this work. Without this clarification, this work only looks like a lab report, with significant gaps towards a research paper.

Author response: We appreciate your comments and the opportunity to explain better this important aspect of our work. As it is well known, a higher number of output voltage levels in a power converter brings better waveform quality, less dv/dt and the consequent reduction in filtering requirements. The compromise is then on the increased component count and practical difficulties in the modulation and control of the converter that might arise in a converter with a higher number of levels. With this in mind, we propose a novel single-phase inverter that is able to produce 4 levels using a topology that is simpler and uses less semiconductor switches than the conventional inverters such as NPC, active-NPC, CHB or flying capacitor.

Author action: We have modified the Introduction in order to justify the research interest in the four-level topology addressed in this paper.

Comment #2: Both the abstract and introduction are a significant lack of quan- titative evidence. Only qualitative descriptions are provided without any numbers to support the state- ment, which is neither informative nor convincing.

Author response: Thanks for pointing out this deficiency in our original manuscript.

Author action: According to your observations, the introduction and the abstract were improved with more quantitative performance indices of the proposed topology and its comparison with conventional multilevel inverters.

Comment #3: The key experimental performance of the proposed inverter is missing in the abstract.

Author action: The abstract now mentions the achievement of validating experimentally the proposed multi-level topology.

Comment #4: Introduction: the authors made a very high-level comparison state- ment of different MLI technologies among cascaded H-Bridge (CHB), the neutral-point clamped (NPC), and the flying capacitor (FC). First, the schematic of each topology is suggested to add for illustration. Second, a more critical assessment of their disadvantages is required, by supplying some numbers. For instance, “NPC its excessive component count limits the expansion of the system to higher number of levels, requiring also complex capacitor voltage-balancing schemes”. Please consider answering the- se questions: How excessive component count? How many components for achieving higher number of levels? Why we need a higher number of levels? What is the optimum number of levels? Similar ques- tions to clarify for FC: FC has the drawback of requiring a high number of large capacitors. How many capacitors and how large are they?

Author response: Thank you for your valuable comments that have helped us to improve this part of our manuscript.

Author action: We have included a new figure with schematic diagrams of the NPC, CHB and FC topologies. We have modified the Introduction and additional effort has been paid to address the relevant questions formulated by the reviewer. Pertinent references have been included in order to support the assessment of known limitations of existing multilevel topologies.

Comment #5: There is a lack of up-to-date literature survey for the modular multilevel converter/inverter part and the modulation strategy - some references are outdated (5-10 years ago). The following paper on the state of the art is recommended to cite. J. Li, “Design and Con- trol Optimisation of a Novel Bypass-embedded Multilevel Multicell Inverter for Hybrid Electric Vehicle Drives,” 2020 IEEE 11th International Symposium on Power Electronics for Distributed Generation Systems (PEDG), Dubrovnik, Croatia, 2020, pp. 382-385, doi: 10.1109/PEDG48541.2020.9244313.

Author response: We appreciate your recommendation of citing this relevant reference to support the literature background of the modulation strategy.

Author action: We have added the reference in section 2.2 of the revised paper.

Comment #6: The authors presented Figure 1 in the Introduction without men- tioning it. It should be placed in section 2.

Author action: Fig. 1 has been relocated to section 2, as suggested by the reviewer.

Comment #7: Figures 10-12: Are these waveform images taken from experimental photos or screenshots of the instrument? They are with extremely low resolution (cannot read it without zooming in) and without notation of which is voltage, and which is current. I suggest to save the data, process them, and replot.

Author response: Thank you for your valuable comments about the quality of Figs. 10-12. These images have been obtained directly from the oscilloscope using the highest resolution available from the instrument. Unfortunately we do not have access to the data so it is impossible to replot the figures. As a solution for the low quality figures, we have enlarged them and cropped the images to focus on the relevant information. We hope that this solution helps improving the presentation of the results.

Author action: Figures 10-12 of the original manuscript have been enlarged and cropped to enable a better visualization. In the revised paper they were also combined in a single figure with multiple parts, following the request from another reviewer.

Comment #8: “dc” should be capitalized throughout the manuscript.
Author action: All the occurrences of “dc”, standing for “direct current” have been capitalized throughout the manuscript. For the sake of consistency, the acronym “ac” is now also capitalized.

Reviewer 4 Report

The authors proposed a four-level single-phase multilevel converter that contained six active switches arranged in a quasi-nested configuration for
single-phase applications. The research idea is well conceived, the results are scientifically sound and the manuscript is well organized and written. Furthermore, the authors have validated their results with simulations and experimentation. I believe it would be a good contribution to electronics. Hence, I would like to accept the manuscript after minor revision of the comments in the next:

1- Provide the bibliographic reference to all the underived base equations used in the manuscript.

2- Merge Figures 10, 11 and 12 as multiple parts of a single Figure as a, b and c.

3- Propose future directions in the conclusion section.

Author Response

Reviewer’s comment: The authors proposed a four-level single-phase multilevel converter that contained six active switches arranged in a quasi-nested configuration for single-phase applications. The research idea is well conceived, the results are scientifically sound and the manuscript is well organized and written. Furthermore, the authors have validated their results with simulations and experimentation. I believe it would be a good contribution to electronics. Hence, I would like to accept the manuscript after minor revision of the comments in the next:

Author response: We are deeply grateful for your positive and encouraging comments. We humbly hope that the revised version of our manuscript meets with your approval.

Comment #1: Provide the bibliographic reference to all the underived base equa- tions used in the manuscript.

Author response: Thank you for your valuable suggestion. We have included references for all equations that are not derived in the manuscript.

Author action: The revised manuscript now contains references for the following equations: (2)-(4), (6), (7) and (11).

Comment #2: Merge Figures 10, 11 and 12 as multiple parts of a single Figure as a, b and c.

Author action: In the revised paper Figs. 10, 11 and 12 have been merged in a single figure, according to the reviewer’s advice.

Comment #3: Propose future directions in the conclusion section. Author response: We appreciate your valuable suggestion.
Author action: The following paragraph has been included in the Conclusion:

“Future research on the proposed MLI will be focused on the development of new control strategies specifically designed for this topology, reactive power control for grid-connected applications and interleaved configuration for enhanced harmonic performance.”

Reviewer 5 Report

The voltage source inverters (VSI) are very useful for various industrial applications. Improving the harmonic spectrum of the VSI output voltage can be obtained by modulating the output voltage at several levels, thus the significance of the multilevel converter is without doubt.

At the beginning, the authors introduce the information about several multilevel inverters such as cascaded H-bridge multilevel inverter (CHB-MLI), the neutral-point clamped (NPC) and the flying capacitor multilevel inverter (FC-MLI) configurations. They precisely discuss the main advantages and disadvantages of the different multilevel inverters’ topologies.

The converter topology, proposed in the paper, is well described, including switching states, implemented modulation scheme and the main control objectives.

The authors show multiple simulation and experimental results for voltage and current waveforms at the different modulation indexes.

I have the following recommendations towards the paper:

  • The application of the principle of output voltage modulation at several levels in single-phase voltage inverts must be justified much better, as the increased number of devices is a serious drawback.
  • The manuscript, without doubt, is relevant. I would recommend to the authors, for greater completeness of their statement, to comment in more detail about the total harmonic distortion (THD) capabilities in the proposed circuit. In addition, the spectrums of the output quantities should be depicted and analyzed.

Author Response

Reviewer’s comment: The voltage source inverters (VSI) are very useful for various industrial applications. Improving the harmonic spectrum of the VSI output voltage can be obtained by modulating the output voltage at several levels, thus the significance of the multilevel converter is without doubt.

At the beginning, the authors introduce the information about several multilevel inverters such as cascaded H-bridge multilevel inverter (CHB-MLI), the neutral-point clamped (NPC) and the flying capacitor multilevel inverter (FC-MLI) configurations. They precisely discuss the main advantages and disadvantages of the different multilevel inverters’ topologies.

The converter topology, proposed in the paper, is well described, including switching states, implemented modulation scheme and the main control objectives.

The authors show multiple simulation and experimental results for voltage and current waveforms at the different modulation indexes.

I have the following recommendations towards the paper:

Author response: We are very grateful for your constructive and encouraging comments. We have carefully studied your comments, and humbly hope the revised version of our manuscript meets with your approval.

Comment #1: The application of the principle of output voltage modulation at several levels in single-phase voltage inverts must be justified much better, as the increased number of devices is a serious drawback.

Author response: Thank you for your valuable suggestion. We fully agree with you that a high number of semiconductor switches can be seen as an important drawback, and that the benefits of single-phase multilevel inverter need to be emphasized.

Author action: Following your advice, we modified the Introduction to highlight the advantages of multilevel waveforms with respect to those from a two-level inverter, which have motivated a great research interest in single-phase applications.

Comment #2: The manuscript, without doubt, is relevant. I would recommend to the authors, for greater completeness of their statement, to comment in more detail about the total harmonic distortion (THD) capabilities in the proposed circuit. In addition, the spectrums of the output quantities should be depicted and analyzed.

Author response: Many thanks for your helpful comments.

Author action: According to your comments, in section 2.2 of the revised manuscript we have included a new figure with the voltage spectrum of the proposed inverter, as well as the corresponding analysis. Additional comments about the THD performance of the proposed topology in comparison with conventional multilevel inverters, shown in the comparative table presented in section 5.1, have been included.

Round 2

Reviewer 1 Report

The author has responded well to most of the comments. Good work.

Author Response

We are sincerely grateful for your detailed and professional review of our work.

Reviewer 2 Report

Authors have made changes in manuscript as per reviewer comments. Now the paper qualifies for publication in MDPI Electronics.
Congratulations!

Author Response

(The authors gave the same response as above.)

Reviewer 3 Report

Most concerns raised from the last round of review have been addressed.
Figures 10-12 are missing notations in the pictures - which curve is for voltage, and which for current? This may not be immediately clear to all readers. I suggest adding denotations in the graphs.

Author Response

We are sincerely grateful for your detailed and professional review of our work. We have included the corresponding notations as suggested.